# Bright, efficient, and stable pure-green hyperfluorescent organic light-emitting diodes by judicious molecular design

Yi-Ting Lee[1,2], Chin-Yiu Chan [3,4] ✉, Nanami Matsuno[5], Shigetada Uemura[5], Susumu Oda[6], Masakazu Kondo[7], Rangani Wathsala Weerasinghe[1], Yanmei Hu[1], Gerardus N. Iswara Lestanto[1], Youichi Tsuchiya [1], Yufang Li [3,4], Takuji Hatakeyama [5] ✉ & Chihaya Adachi [1,8] ✉

To fulfill ultra-high-definition display, efficient and bright green organic light-emitting diodes with Commission Internationale de l'Éclairage y-coordinate ≥ 0.7 are required. Although there are some preceding reports of highly efficient devices based on pure-green multi-resonance emitters, the efficiency rolloff and device stabilities for those pure-green devices are still unsatisfactory. Herein, we report the rational design of two pure-green multi-resonance emitters for achieving highly stable and efficient pure-green devices with $CIE_{x,y}$s that are close to the NTSC and BT. 2020 standards. In this study, our thermally activated delayed fluorescence OLEDs based on two pure-green multi-resonance emitters result in $CIE_y$ up to 0.74. In hyperfluorescent device architecture, the $CIE_x$s further meet the x-coordinate requirements, i.e., NTSC (0.21) and BT. 2020 (0.17), while keeping their $CIE_y$s ~ 0.7. Most importantly, hyperfluorescent devices display the high maximum external quantum efficiencies of over 25% and maximum luminance of over $10^5$ cd m$^{-2}$ with suppressed rolloffs (external quantum efficiency of ~20% at $10^4$ cd m$^{-2}$) and long device stabilities with $LT_{95}$s of ~ 600 h.

To alleviate the supply threat of strategic precious metals and promote environmental sustainability, organic light-emitting diodes (OLEDs) based on thermally activated delayed fluorescent (TADF) emitters, which are precious metal-free organic materials, have attracted intense attention recently[1–4]. However, the present TADF OLEDs are still insufficient to achieve the compatibilities of high efficiency, high color purity, high brightness, and high device stability simultaneously[5–11].

OLEDs based on metal-free donor-acceptor (DA)-type TADF emitters realizing 100% internal quantum efficiencies were first demonstrated by Adachi and co-workers in 2012[2]. Since then, blue,

green, and red TADF OLEDs with high external quantum efficiencies (EQEs) of up to 40% have been reported via the rational design of donor and acceptor motifs of the TADF emitters[12–15]. However, the charge-transfer (CT) emission nature of DA-type emitters always results in broadband emission, which is difficult to satisfy the green color gamut standard with y-coordinate over 0.7[16]. The Commission Internationale de l'Éclairage coordinates ($CIE_{x,y}$) for National Television Standards Committee (NTSC) standard in green color are (0.21, 0.71)[16]. Meanwhile, the NTSC standard in green has been gradually advanced to BT. 2020 standard, in which $CIE_{x,y}$ is required to be (0.17,

[1]Center for Organic Photonics and Electronics Research (OPERA), Kyushu University, Motooka, Nishi, Fukuoka, Japan. [2]Department of Chemistry, Soochow University, Taipei, ROC, Taiwan. [3]Department of Materials Science and Engineering, City University of Hong Kong, Tat Chee Avenue, Kowloon, Hong Kong SAR, China. [4]Department of Chemistry, City University of Hong Kong, Tat Chee Avenue, Kowloon, Hong Kong SAR, China. [5]Department of Chemistry, Graduate School of Science, Kyoto University, Sakyo-ku, Kyoto, Japan. [6]Department of Applied Chemistry, Graduate School of Science and Engineering, Toyo University, Kawagoe, Saitama, Japan. [7]JNC Co., Ltd. 5-1 Goikaigan, Ichihara, Chiba, Japan. [8]International Institute for Carbon Neutral Energy Research (WPI-I2CNER), Kyushu University, Nishi, Fukuoka, Japan. ✉e-mail: chinychan2@cityu.edu.hk; hatake@kuchem.kyoto-u.ac.jp; adachi@cstf.kyushu-u.ac.jp

0.80) aimed at ultra-high definition (UHD) display[16]. To solve the color purity issue, in 2016, Hatakeyama and co-workers first developed a class of pure-blue TADF emitters, namely multi-resonance emitter (MRE), which features the characteristics of narrowband emission and high photoluminescence quantum yields (PLQYs)[9]. Since then, full-color MRE-based TADF OLEDs have attracted a lot of attention[17–25]. Nowadays, except for pure-blue and pure-red MREs, there have been several reports on the design and synthesis of pure-green MREs, which result in high-performance pure-green OLEDs (Supplementary Table 1). However, pure-green MRE-based OLEDs with $CIE_y \geq 0.7$ have seldom been reported in the literature. Until recently, there have been a few reports on the design and synthesis of pure-green MREs for achieving high-efficiency devices with $CIE_y s \geq 0.7$[26–28]. Nonetheless, the device stabilities and efficiency rolloff issue for pure-green devices with $CIE_y s \geq 0.7$ are always unsatisfactory[26–28]. On the other hand, when targeting stable pure-green OLEDs with a $CIE_y \geq 0.7$, $CIE_x$ is also equally important to be satisfied for NTSC standard ($CIE_x = 0.21$) or BT. 2020 standard ($CIE_x = 0.17$), which is seldom given attention and is usually compromised in pure-green OLEDs[16]. Therefore, there is a strong demand for the design and synthesis of pure-green MREs to obtain bright, efficient, and stable pure-green OLEDs, while both x- and y-coordinates of the electroluminescence (EL) meet the NTSC or BT. 2020 standard.

In this article, we report the design and synthesis of two pure-green MREs, namely ω-DABNA-M and ω-DABNA-PH (Fig. 1). By extending π−conjugation or increasing donor strength, the emissions of ω-DABNA-M and ω-DABNA-PH have been significantly redshifted when compared to early reported MRE (ω-DABNA)[27]. TADF OLEDs based on these three pure-green MREs not only result in $CIE_y s \geq 0.7$ but also better $CIE_x s$. Furthermore, when hyperfluorescence (HF) OLED architectures are employed, the $CIE_x s$ meet the requirements of x-coordinate for NTSC standard (0.21) and BT 2020 standard (0.17) while keeping their $CIE_y s \sim 0.7$. Additionally, HF OLEDs based on ω-DABNA, ω-DABNA-M, and ω-DABNA-PH displayed the maximum EQEs of 28.0%, 28.3%, and 27.3%, respectively. Efficiency rolloff issues, which

are commonly observed in MRE-only OLEDs, are greatly suppressed in all HF OLEDs. With judicious molecular design of MREs, our HF devices not only can achieve high maximum brightness of over $10^5$ cd m$^{-2}$, but also can maintain high EQEs of ~20% even at high brightness of $10^4$ cd m$^{-2}$, which are important for UHD display. Most importantly, by controlling the dopant concentration of MREs and utilizing suitable TADF assistant dopant in HF OLEDs, the pure-green HF OLEDs exhibit long device stabilities with $LT_{95}s$ of ca. 600 h at an initial luminance of $10^3$ cd m$^{-2}$.

## Results

### Molecular design and photophysical properties

The early reported pure-green MRE (ω-DABNA) emitted photoluminescence (PL) at 509 nm with a doping concentration of 1 wt% in a poly(methyl methacrylate) (PMMA) film. The OLED based on ω-DABNA gave rise to an EL maximum of 512 nm[27]. However, the corresponding $CIE_{x,y}$ (0.13, 0.73) and device stability are still unsatisfactory. To have a better $CIE_x$ and $CIE_y$, the emission should be redshifted. Moreover, the efficiency rolloff and brightness of the previously reported devices should also be improved. Thus, chemical modification on ω-DABNA is highly desired. The first approach is to increase the donor strength by the alkyl substitution (ω-DABNA-M). The emission is expected to be redshifted; however, it is anticipated that alkyl substitution, which always lowers the carrier transporting properties in the emitting layer, will affect the device performance. Therefore, instead of the alkyl substitution, the second approach is to extend the π−conjugation by phenyl group (ω-DABNA-PH). It is envisaged that the positive mesomeric effect by phenyl substitution is more effective to redshift the emission of MRE and achieve better device performance, when compared to the positive inductive effect by methyl substitution.

### Synthesis and computational simulations

ω-DABNA-M and ω-DABNA-PH were synthesized according to the reported ω-DABNA, which involved sequential two-step Friedel-Craft borylations (Supplementary Note 1 and Supplementary Fig. 1). Both ω-

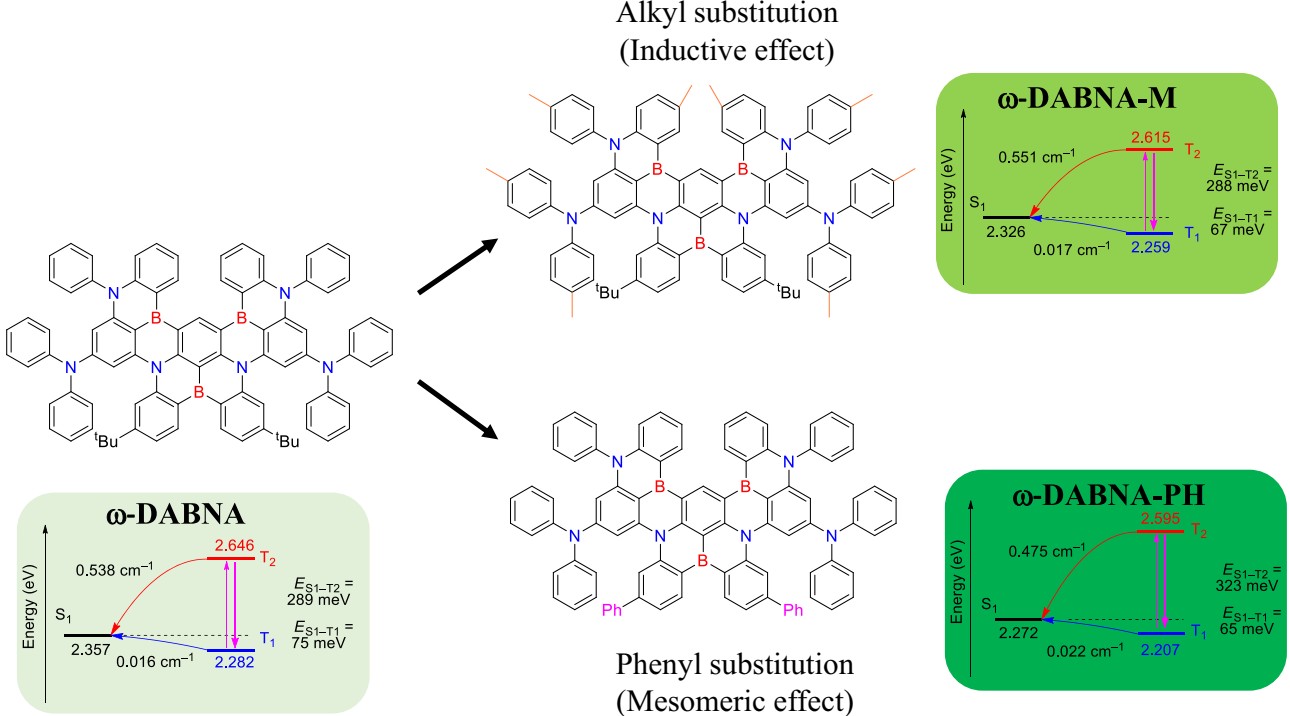

**Fig. 1 | Design strategy and density-functional theory calculation.** Molecular structures of ω-DABNA, ω-DABNA-M, and ω-DABNA-PH with their corresponding strategy for achieving better $CIE_{x,y}$ with TDA-B2PLYP(cx = 0.40, cc = 0.23)/cc-pVDZ//M06-2X/6-31 G(d) level of theory.

DABNA-M and ω-DABNA-PH were obtained with high product yields. The intermediates and final compounds were well characterized by NMR and mass spectrometry (Supplementary Figs. 2–15). Time-dependent density-functional theory (TD-DFT) calculation was employed to estimate and compare the highest occupied molecular orbital (HOMO) and lowest unoccupied molecular orbital (LUMO) levels of three green MREs (Supplementary Note 2, Supplementary Tables 2 and 3, and Supplementary Figs. 16-18). It was found that both HOMO and LUMO levels of ω-DABNA-M (−4.47 and −1.39 eV, respectively) were significantly affected by the inductive effect of methyl groups, in which the HOMO and LUMO were more high-lying when comparing to those of the parent ω-DABNA (−4.61 and −1.50 eV, respectively). The extent of the change in HOMO was higher than that in LUMO, hence resulting in a narrower energy gap and a redshifted emission; however, in ω-DABNA-PH, the LUMO (−1.66 eV) was greatly stabilized by the mesomeric effect of phenyl groups, whereas HOMO was only slightly affected (−4.67 eV). It was found that the electron distribution in LUMO is extended to the phenyl group that insinuates ω-DABNA-PH having more CT properties. The LUMO was found to be more low-lying when compared to that of ω-DABNA, hence leading to a narrower energy gap and a more redshifted emission. We also performed TDA-B2PLYP(cx = 0.40, cc = 0.23)/cc-pVDZ//M06-2X/6-31 G(d) level of theory to estimate the excited properties of three MREs (Supplementary Fig. 19)[29]. The calculated singlet excited state ($S_1$) energies of ω-DABNA, ω-DABNA-M, and ω-DABNA-PH were 2.357, 2.326, and 2.272 eV, respectively, which were consistent with the TD-TDF and indicated the tendency of more redshifted emissions. The singlet-triplet energy gap ($\Delta E_{S1-T1}$) of ω-DABNA, ω-DABNA-M and ω-DABNA-PH were found to be 75, 67 and 65 meV, respectively. The reduced $\Delta E_{S1-T1}$s in ω-DABNA-M and ω-DABNA-PH were expected to show enhanced TADF properties when compared to ω-DABNA, hence leading to improved device performance. Notably, the spin-orbital coupling (SOC) matrix elements (SOC = $\langle S_n | \hat{H}_{SOC} | T_n \rangle$) of ω-DABNA-M ($S_1 - T_1$, 0.017 cm$^{-1}$; $S_1 - T_2$, 0.551 cm$^{-1}$) were smaller than that of parent ω-DABNA ($S_1 - T_1$, 0.016 cm$^{-1}$; $S_1 - T_2$, 0.538 cm$^{-1}$). Contrary, ω-DABNA-PH showed much larger SOC matrix elements ($S_1 - T_1$, 0.022 cm$^{-1}$; $S_1 - T_2$, 0.475 cm$^{-1}$), which could be attributed to the small $\Delta E_{S1-T1}$ and more distorted molecular structure by the bulky phenyl groups. The cartesian coordinates of ω-DABNA, ω-DABNA-M and ω-DABNA-PH were tabulated in Supplementary Tables 4–6.

## Photophysical properties in thin films

After the successful synthesis of ω-DABNA-M and ω-DABNA-PH, we tested the photophysics by doping 1 wt% of ω-DABNA-M and ω-DABNA-PH in a PMMA film, which was similar to ω-DABNA (Table 1 and Supplementary Figs. 20–27)[26]. Both ω-DABNA-M and ω-DABNA-PH showed slightly redshifted absorption peaks of 500 and 502 nm, respectively, when compared to the parent ω-DABNA. On the other hand, the emission maxima of ω-DABNA-M and ω-DABNA-PH were found to be 514 and 512 nm, respectively, which were slightly redshifted than that of ω-DABNA (509 nm). Two green MREs displayed characteristic narrowband emissions with corresponding full-width at half-maximums (FWHMs) of 23 nm (ω-DABNA-M) and 28 nm (ω-DABNA-PH). All MREs exhibited high PLQYs of around 87%, which is

expected to result in high device efficiency. The delayed lifetime ($\tau_d$) of ω-DABNA was 8.95 μs, while $\tau_d$s of ω-DABNA-M and ω-DABNA-PH were found to be greatly shortened. The $k_{RISC}$ of ω-DABNA was moderate (1.2 ×10$^5$ s$^{-1}$); however, ω-DABNA-M and ω-DABNA-PH showed a slightly higher $k_{RISC}$ of 2.1 ×10$^5$ s$^{-1}$ and 2.2 × 10$^5$ s$^{-1}$, respectively. The shorter $\tau_d$s and higher $k_{RISC}$s of ω-DABNA-M and ω-DABNA-PH would be originated from smaller $\Delta E_{S1-T1}$s and higher SOC matrix elements. All the rate constants were calculated and tabulated in Table 1 with more details in Supplementary Notes 3-5 and Supplementary Figs. 21–23. We also tested the transient decay profiles of HF films with a sky-blue TADF material as an assistant dopant (Supplementary Figs. 24-27). 3Cz2DPhCzBN was selected as the assistant dopant since the $T_1$ of 3Cz2DPhCzBN[7] (2.72 eV) is higher than the $T_1$s of ω-DABNA-M, ω-DABNA-M, and ω-DABNA-PH and the suitable spectrum overlap between the emission spectrum of 3Cz2DPhCzBN and the absorption spectrum of the MREs. All ω-DABNA, ω-DABNA-M, and ω-DABNA-PH HF films display fast delayed lifetimes of 0.99 μs, 0.78 μs, and 0.75 μs, respectively, indicating fast reductions of triplet excitons by HF strategy. The ω-DABNA-PH HF film is the most effective. Also, high PLQYs of 58%, 69%, and 74% are found in ω-DABNA, ω-DABNA-M, and ω-DABNA-PH HF doped films, respectively.

## Electrochemical study

The electrochemical properties of MREs were investigated by cyclic voltammetry measurement (Supplementary Fig. 28). From the oxidative scan of cyclic voltammograms, the HOMO energy levels of ω-DABNA, ω-DABNA-M, and ω-DABNA-PH were found to be −5.26, −5.21, and −5.24 eV, respectively, which were determined from the oxidative scan of cyclic voltammograms. From the HOMO energy level and optical energy gap, the LUMO energy levels of ω-DABNA, ω-DABNA-M and ω-DABNA-PH were estimated from the HOMO energy level and optical energy gap, which were found to be −2.82, −2.79, and −2.82 eV, respectively. The experimental HOMO and LUMO energy levels are well-matched to the calculated values. It was found that the introduction of methyl groups on ω-DABNA-M results in an inductive effect, which raises the energy of both HOMO and LUMO. On the other hand, the mesomeric effect of phenyl substitution on ω-DABNA-PH maintained the LUMO energy level, but with a shallower HOMO energy level, hence resulting in a redshifted emission.

## Device performance

To examine the device performance based on ω-DABNA, ω-DABNA-M, and ω-DABNA-PH, the following configuration: indium-tin oxide (ITO)-coated glass (50 nm)/ NPD (40 nm)/ TCTA (15 nm)/ mCP (15 nm)/ DOBNA-Ph: 0.5 wt% MRE (20 nm)/ 3,4-2CzBN (10 nm)/ BPy-TP2 (20 nm)/ LiF (0.8 nm)/Al (100 nm) was used as TADF OLEDs, where *N*,*N'*-di(1-naphthyl)-*N*,*N'*-diphenyl-(1,1'-biphenyl)-4,4'-diamine (NPD) is the hole-injection layer, tris(4-carbazoyl-9-ylphenyl)amine (TCTA) is the hole-transporting layer, 1,3-bis(*N*-carbazolyl)benzene (mCP) is the electron-blocking layer, 3,11-diphenyl-5,9-dioxa-13b-boranaphtho[3,2,1-*de*]anthracene (DOBNA-Ph) is the host, 3,4-di(9*H*-carbazol-9-yl) benzonitrile (3,4-2CzBN) is the hole-blocking layer, 2-(9,9'-spirobi[fluoren]-3-yl)-4,6-diphenyl-1,3,5-triazine (BPy-TP2) is the electron-transporting layer, and lithium fluoride (LiF) and Al are the

**Table 1 | Basic photophysical parameters of ω-DABNA, ω-DABNA-M, and ω-DABNA-PH**

| Compounds | In PMMA (1 wt%) | | | | | | | | | | |
|---|---|---|---|---|---|---|---|---|---|---|---|
| | $\lambda_{abs}$ (nm) | $\lambda_{max}$ (nm) | Stock shift (nm) | FWHM$^a$ (nm) | $\Delta E_{ST}^b$ (eV) | $\Phi_{Ar}^c$ (%) | $\Phi_p^d$ (%) | $\Phi_d^e$ (%) | $\tau_p^f$ (ns) | $\tau_d^g$ (μs) | $k_{RISC}$ (10$^5$ s$^{-1}$) |
| ω-DABNA | 495 | 509 | 14 (556 cm$^{-1}$) | 22 (665 cm$^{-1}$) | 0.013 | 0.870 | 0.819 | 0.051 | 5.92 | 8.95 | 1.2 |
| ω-DABNA-M | 500 | 514 | 14 (545 cm$^{-1}$) | 23 (882 cm$^{-1}$) | 0.010 | 0.874 | 0.838 | 0.036 | 5.60 | 5.04 | 2.1 |
| ω-DABNA-PH | 502 | 512 | 10 (465 cm$^{-1}$) | 28 (1072 cm$^{-1}$) | 0.018 | 0.865 | 0.830 | 0.035 | 4.62 | 4.70 | 2.2 |

$^a$full-width at half-maximum; $^b$singlet-triplet energy gap; $^c$PLQY in argon atmosphere; $^d$prompt component; $^e$delayed component; $^f$prompt lifetime; $^g$delayed lifetime.

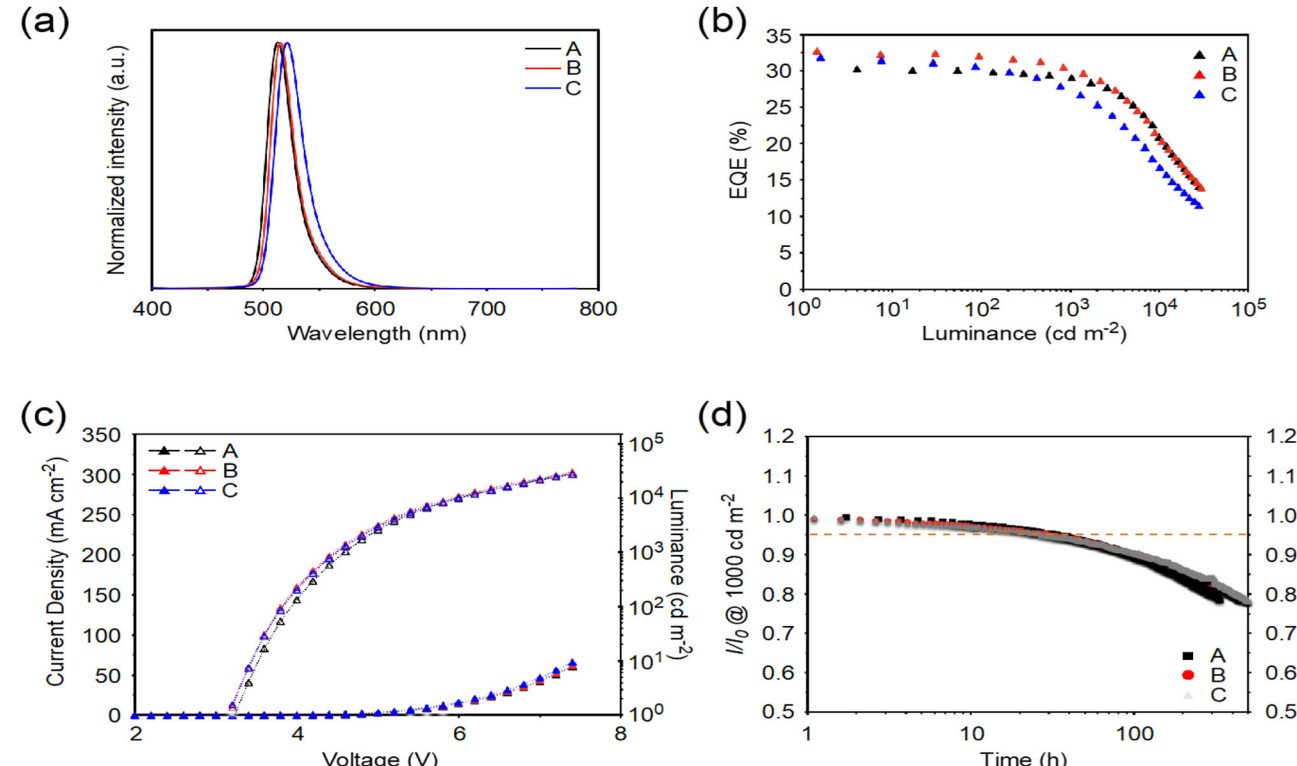

**Fig. 2 | Device performance of TADF devices A−C. a** EL spectra; **b** EQE versus luminance curves; **c** current density-voltage-luminance curves, and **d** device stability (at an initial luminance of 1000 cd m⁻²).

electron injection and cathode layers, respectively (Supplementary Figs. 29 and 30). With this device structure, the OLEDs **A** − **C** based on ω-DABNA, ω-DABNA-M, and ω-DABNA-PH were fabricated, respectively. The device data is summarized in Fig. 2 and Table 2. Device **A** as a reference displayed pure-green EL with an emission maximum of 512 nm and an FWHM of 25 nm, corresponding to $CIE_{x,y}$ of (0.13, 0.73), and achieved a maximum EQE of 30.8% and maintained a high value of 29.2% at 1000 cd m⁻². To enhance the donor strength for a redshifted emission, the methyl substitution on ω-DABNA-M has been employed, which slightly redshifts the emission while keeping a narrow FWHM. Device **B** based on ω-DABNA-M emitted pure-green color at 515 nm with an FWHM of 25 nm, corresponding to better $CIE_{x,y}$ of (0.15, 0.74), and exhibited the highest maximum EQE of 32.7%. On the other hand, by replacing the *tert*-butyl group with the phenyl group, the extended π−conjugation in ω-DABNA-PH resulted in redshifted emissions both in the solution and the doped film. Consistently, device **C** based on ω-DABNA-PH displayed the reddest EL at 521 nm with an FWHM of 30 nm, $CIE_{x,y}$ of (0.19, 0.74), and a high maximum EQE of 32.7%. It should be noted that the extended π−conjugation in ω-DABNA-PH significantly redshifted the emission; however, the FWHM was slightly broadened as well, hence resulting in a large increase in $CIE_x$ to 0.19. Devices **A** − **C** all showed decent device stabilities with $LT_{95}$s (95% of the initial luminance) of 34, 24, and 27 h, respectively. However, the resulting device stabilities were still unsatisfactory.

In addition to high efficiency, device stability is extremely important from the aspect of practical applications. As a result, an HF OLED architecture was introduced, where 3Cz2DPhCzBN[7] was used as a sky-blue TADF assistant dopant and the MREs were used as the terminal emitter. HF devices were prepared with the following device structure: ITO-coated glass (100 nm)/HAT-CN (10 nm)/ Tris-PCz (30 nm)/mCBP (5 nm)/mCBP: 20 wt% 3Cz2DPhCzBN: 0.5 wt% MRE (30 nm)/SF3-TRZ (10 nm)/SF3-TRZ: 30 wt% Liq (30 nm)/Liq (2 nm)/ Al (100 nm) was used as a HF device, where 1,4,5,8,9,11-hexaazatriphenyl-enehexacarbonitrile (HAT-CN) is the hole-injection layer, 9-phenyl-3,6-

bis(9-phenyl-9*H*carbazol-3-yl)-9*H*-carbazole (tris-PCz) is the hole-transporting layer, mCBP was used for exciton-blocking and host layers, 3Cz2DPhCzBN is a TADF assistant dopant, 2-(9,9'-spirobi[fluoren]-3-yl)-4,6-diphenyl-1,3,5-triazine (SF3-TRZ) is the electron-transporting layer, and 8-hydroxyquinolinolato-lithium (Liq) and Al are the electron injection and cathode layers, respectively (Supplementary Figs. 29 and 30). The reference device based on 20 wt% 3Cz2DPhCzBN was also fabricated with the same device structure for a better comparison (Supplementary Fig. 31). In a 20 wt% 3Cz2DPhCzBN-based reference device (**Ref-1**), a maximum EQE of 23 % with an EL at 498 nm was achieved, corresponding to $CIE_{x,y}$ of (0.22, 0.48). The device showed an $LT_{95}$ of 89 h at an initial luminescence of 1000 cd cm⁻². On the other hand, HF OLED devices **D** − **F** were fabricated based on ω-DABNA, ω-DABNA-M, and ω-DABNA-PH, respectively. To minimize the direct formation of triplet excitons on MRE, the doping concentrations of MREs were first kept to be 0.5 wt% in devices **D** − **F**, so that all excitons were generated on TADF assistant dopant first and then were transferred to MRE via Förster resonance energy transfer (FRET)[30]. Moreover, the $k_{RISC}$ of MRE is usually lower than that of the assistant dopant. Therefore, with an increase in MRE concentration, more triplet excitons are directly formed on MRE, hence resulting in a larger efficiency rolloff. The corresponding device characteristics of devices **D** − **F** are depicted in Fig. 3 and tabulated in Table 2. Device **D** emitted at 513 nm, which is similar to that of device **A**, indicating that the emission originates from ω-DABNA. Although the spectral overlap between the emission of 3Cz2DPhCzBN and the absorption of ω-DABNA was good, the low dopant concentration of ω-DABNA (0.5 wt%) resulted in an incomplete FRET between them. As a result, there was an emission shoulder at <500 nm originating from 3Cz2DPhCzBN. The $CIE_{x,y}$ of device **D** was found to be (0.15, 0.64), and achieved a maximum EQE of 28.0% and kept a high EQE of 19.5% at 10⁴ cd m⁻².

On the other hand, devices **E** and **F** based on ω-DABNA-M and ω-DABNA-PH emitted redshifted pure-green ELs at 515 and 521 nm with the FWHMs of 25 and 30 nm, respectively, which led to corresponding

**Table 2 | Summary of device characteristics of devices A–K and reference devices**

| Device | Dopant | $V_{on}^{a}$ (V) | $CE^{b}$ (cd A⁻¹) | $PE^{c}$ (lm W⁻¹) | $EQE^{d}$ (%) | $L_{max}^{e}$ (cd m⁻²) | $\lambda_{EL}^{f}$ (nm) | $FWHM^{g}$ (nm) | CIE (x, y)$^{h}$ | $LT_{95}^{i}$ (hour) @ 1000 cd m⁻² |
|---|---|---|---|---|---|---|---|---|---|---|
| A (reference) | 0.5 wt% ω-DABNA | 3.0 | 101.9 | 106.2 | 30.8/ 30.4/ 29.2/– | – | 512 | 25 | (0.13, 0.73) | 34 |
| B | 0.5 wt% ω-DABNA-M | 3.2 | 114.1 | 114.5 | 32.7/ 32.0/ 30.3/– | – | 515 | 25 | (0.15, 0.74) | 24 |
| C | 0.5 wt% ω-DABNA-PH | 3.1 | 123.6 | 124.8 | 31.8/ 30.5/ 27.4/– | – | 521 | 30 | (0.19, 0.74) | 27 |
| D | 0.5 wt% ω-DABNA: 20 wt% 3Cz2DPhCzBN | 3.0 | 82.5 | 86.4 | 28.0/ 25.1/ 23.9/ 19.5 | 92351 | 511 | 24 | (0.15, 0.64) | 188 |
| E | 0.5 wt% ω-DABNA-M: 20 wt% 3Cz2DPhCzBN | 3.0 | 86.4 | 90.5 | 28.3/ 26.8/ 25.0/ 19.4 | 84098 | 515 | 25 | (0.18, 0.65) | 230 |
| F | 0.5 wt% ω-DABNA-PH: 20 wt% 3Cz2DPhCzBN | 3.0 | 89.6 | 93.8 | 27.3/ 25.6/ 24.6/ 21.1 | 134631 | 521 | 30 | (0.20, 0.66) | 205 |
| G | 1 wt% ω-DABNA: 20 wt% 3Cz2DPhCzBN | 3.5 | 76.6 | 75.2 | 26.8/ 22.3/ 21.2/ 16.8 | 51048 | 513 | 23 | (0.14, 0.69) | 143 |
| H | 1 wt% ω-DABNA-M: 20 wt% 3Cz2DPhCzBN | 3.0 | 94.2 | 98.7 | 28.2/ 25.3/ 22.6/ 16.6 | 72622 | 517 | 23 | (0.17, 0.70) | 110 |
| I | 1 wt% ω-DABNA-PH: 20 wt% 3Cz2DPhCzBN | 3.0 | 94.2 | 98.6 | 26.5/ 24.9/ 23.1/ 19.6 | 155964 | 522 | 29 | (0.21, 0.69) | 109 |
| J | 1 wt% ω-DABNA-M: 20 wt% 4CzIPN | 2.8 | 60.0 | 67.3 | 19.1/ 15.8/ 12.0/ 7.3 | 19302 | 517 | 26 | (0.27, 0.66) | 63 |
| K | 1 wt% ω-DABNA-PH: 20 wt% 4CzIPN | 2.6 | 89.0 | 107.5 | 24.4/ 21.7/ 18.5/ 12.6 | 138483 | 522 | 30 | (0.24, 0.70) | 580 |
| Ref_1 | 20 wt% 3Cz2DPhCzBN | 3.2 | 62.1 | 61.0 | 23.0/ 22.4/ 21.7/ 16.3 | 56290 | 498 | – | (0.22, 0.48) | 89 |
| Ref_2 | 20 wt% 4CzIPN | 3.0 | 71.5 | 74.9 | 21.1/ 20.5/ 19.2/ 16.0 | 139366 | 528 | – | (0.33, 0.60) | 403 |

aturn-on voltage at 1 cd m⁻²; bmaximum current efficiency; cmaximum power efficiency; dvalues at 1, 10², 10³ and 10⁴ cd m⁻²; emaximum luminance at 12 V; felectroluminescence maximum at 10³ cd m⁻²; gfull-width at half-maximum; hvalue at 10³ cd m⁻²; iat an initial luminance of 10³ cd m⁻².

CIE$_{x,y}$s of (0.18, 0.65) and (0.20, 0.66). The redshifted emissions of ω-DABNA-M and ω-DABNA-PH resulted in better CIE$_{x}$s and CIE$_{y}$s in devices **E** and **F**, when compared to that of device **D**. Moreover, devices **E** and **F** also showed maximum EQEs of 28.3 and 27.3%, respectively, and kept high efficiencies of 19.4% and 21.1% at $10^4$ cd m⁻². It was found that the efficiency rolloff is greatly suppressed in device **F** with phenyl-substituted ω-DABNA-PH. When comparing the efficiency at 1 and 1000 cd m⁻² in devices **D** – **F**, devices **D**, **E**, and **F** showed efficiency rolloffs of 14.6 %, 11.7%, and 9.8%, respectively. Since the major dopant in HF devices is the TADF assistant dopant, the carrier transporting properties in HF devices mainly depend on the TADF assistant dopant, instead of MREs. Device **F** was able to achieve a high brightness of 134631 cd m⁻² at 12 V, which was much higher than that of devices **D** (92351 cd m⁻²) and **E** (84098 cd m⁻²). The enhanced maximum brightness of device **F** might be attributed to the improved conductivity of phenyl-modified ω-DABNA-PH. Here, the device stabilities were significantly enhanced (6-10 folds) in devices **D** – **F** when compared to those in devices **A** – **C**. Devices **D** – **F** showed LT$_{95}$s of 188, 230, and 205 h, respectively, at an initial luminance of $10^3$ cd m⁻².

Transient EL measurements (TrELs) were performed on device **E** to understand the exciton dynamic (Supplementary Fig. 32). Without applying a negative voltage at t = 0, the TrEL of device **E** displayed a long emission lifetime. Upon applying negative voltage (from − 2 to −10 V) at t = 0, the intensity of the emission spike at time = 0 was gradually increased. Here, the emission spike can be originated from the excitons formed by de-trapping the trapped holes by ω-DABNA-M, since ω-DABNA-M has the shallowest HOMO levels compared with others. The higher the emission intensity of the spike, the higher the number of hole traps. Furthermore, the device lifetime of EL is extended when fewer hole traps are present in the device.

Because of the incomplete FRET, the resulting CIE$_{x,y}$s in devices **D** – **F** were not good enough. To balance the importance of device stability and color purity that is needed for NTSC and BT. 2020 standards, devices **G** – **I** were fabricated with an increased dopant concentration (1 wt%) of ω-DABNA, ω-DABNA-M, and ω-DABNA-PH, respectively (Fig. 4). By increasing the dopant concentration of the terminal emitter, the FRET became efficient, hence reducing the emission from 3Cz2DPhCzBN. Devices **G** – **I** not only resulted in the high maximum EQEs of 26.8%, 28.2%, and 26.5%, respectively, but also better CIE$_{x,y}$s of (0.14, 0.69), (0.17, 0.70), and (0.21, 0.69), respectively. The efficiency roll-off is significantly minimized in device **I** with ω-DABNA-PH, in which EQEs of 16.8%, 16.6%, and 19.6% were achieved at $10^4$ cd m⁻² in devices **G** – **I**, respectively. In contrast to devices **D** – **F**, it was found that the efficiency rolloff was highly dependent on the k$_{RISC}$ of corresponding MREs in devices **G** – **I**. The higher k$_{RISC}$ of ω-DABNA-PH in device **I** resulted in the least efficiency rolloff. Similar to device **F**, device **I** showed an ultra high maximum brightness of 155964 cd m⁻² at 12 V. The stabilities of devices **G** – **I** were also measured and resulted in the achievement of LT$_{95}$s of 143, 110, and 109 h, respectively. The higher dopant concentration (1 wt%) of ω-DABNA-M in device **H** (compared to device **E**, 0.5 wt%) increased the number of hole traps, which was confirmed by TrEL. In device **H**, the higher dopant concentration of ω-DABNA-M led to an increase in hole traps, which resulted in a higher intensity of the emission spike at t = 0 in TrEL (Supplementary Fig. 32). Hence, device **H** resulted in reduced device stability when compared to that of device **E**. Despite the reduced device stabilities upon increasing the dopant concentration of MRE, the CIE$_{x,y}$s achieved in devices **H** (0.17, 0.70) and **I** (0.21, 0.69) were very close to the BT.2020 and NTSC standards, respectively. The changes of CIE$_{x,y}$s in devices **D** - **I** with respect to luminance have been summarized in Supplementary Table 7 and Supplementary Figs. 33-35.

To further enhance the device stability without losing the color purity, devices **J** and **K** were fabricated with the same device structure to that in devices **H** and **I**, respectively (Supplementary

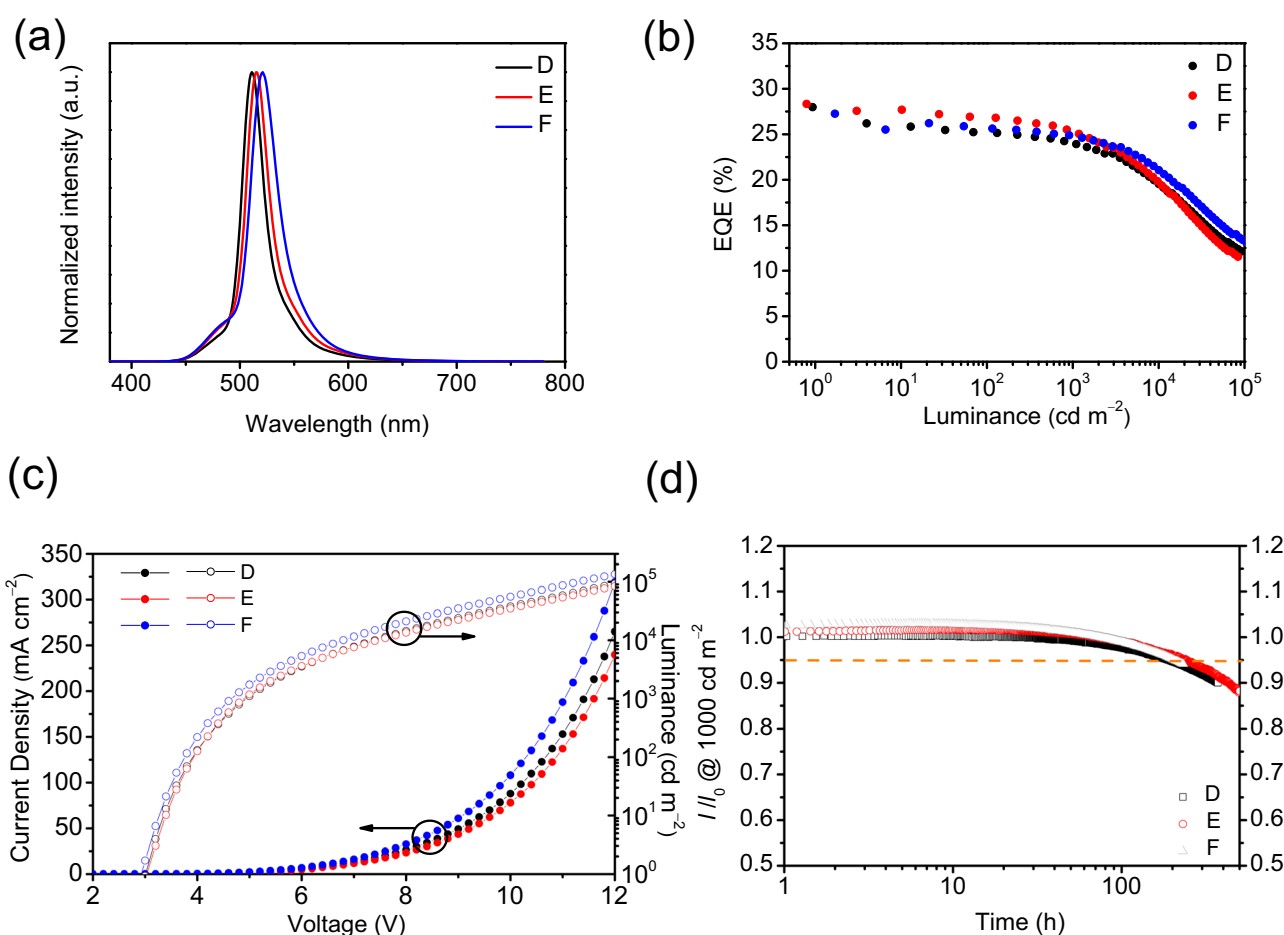

**Fig. 3 | Device performance of HF devices D–F. a** EL spectra; **b** EQE versus luminance curves; **c** current density-voltage-luminance curves and **d** Device stability (at an initial luminance of 1000 cd m⁻²).

Figs. 37 and 38); however, 3Cz2DPhCzBN was replaced by green TADF emitter, 4CzIPN[2]. Again, the reference device based on 20 wt% 4CzIPN (**Ref-2**) was also fabricated with the same device structure for a better comparison (Supplementary Fig. 39). In a 20 wt% 4CzIPN-based reference device, a maximum EQE of 21.1 % with an EL at 528 nm was achieved, corresponding to $CIE_{x,y}$ of (0.33, 0.60). The device showed an extraordinary $LT_{95}$ of 403 h at an initial luminescence of 1000 cd cm⁻². In HF OLED, device **J** resulted in a slightly lower maximum EQE of 19.9%, when compared to that of **Ref-2**. Although the FWHM of the EL spectrum was improved to 26 nm with $CIE_{x,y}$ of (0.27, 0.66), the device stability of device **J** was worse than **Ref-2**. 4CzIPN possesses a deeper HOMO level than that of 3Cz2DPhCzBN. The poorer device stability would be originated from HOMO traps, which is due to a mismatch of HOMO levels between 4CzIPN and ω-DABNA-M. In contrast to device **J**, device **K** displayed a higher maximum EQE of 24.4%, and the EL maximum of 522 nm, which was comparable to that of device **I**, together with a narrow FWHM of 30 nm. Device **K** showed a similar $CIE_{x,y}$ of (0.24, 0.70) to that of device **I** (0.21,0.69). However, the device stability was extraordinarily extended with an $LT_{95}$ of 580 h, a 5-fold increase compared to device **I**. In general, the device stability of the HF device is highly dependent on the stability of the TADF assistant dopant. Further, the degree of overlap between the absorption spectrum of the terminal emitter and the emission spectrum of the assistant dopant and the extinction coefficient of the terminal emission in the overlap range will also affect the device lifetime. The changes of $CIE_{x,y}$s in devices **J** and **K** with respect to luminance have been summarized in Supplementary Table 7 and Supplementary Fig. 36,

while the changes of EL spectra in devices **E** – **K** with respect to time have been depicted in Supplementary Fig. 40.

In summary, pure-green MREs, ω-DABNA-M and ω-DABNA-PH, have been synthesized and devices based on ω-DABNA-M and ω-DABNA-PH have been fabricated. Devices **A** – **C** demonstrated highly efficient pure-green TADF OLEDs. With the help of the HF strategy, devices **D** – **F** demonstrated highly bright, stable and efficient pure-green HF devices. Further tuning the dopant concentration of MREs, device **G** – **K** showed optimized device characteristics. Alkyl substituents are commonly used to construct MREs for better solubility, but it has been found that the phenyl substitution to alkyl groups not only effectively red-shifts the emission of MREs, but also significantly suppresses the efficiency rolloffs and improves the brightness in HF OLEDs. Also, it has been found that the device stability is highly reliant on the TADF assistant dopant. These findings give an insight into the design of efficient MREs towards high-performance and stable HF-OLED. It is also believed that the ultra-bright pure-green HF OLEDs presented here will advance the development of HF technology for AR/VR applications that require OLEDs with high brightness, efficiency, stability and color purity.

## Methods

### Photo-physical measurements

Toluene solutions containing MREs (10⁻⁵ M) were prepared to investigate their absorption and PL characteristics in the solution state. The thin-film samples were deposited on quartz glass substrates by vacuum evaporation to study their exciton confinement properties in the film state. Ultraviolet–visible absorption (UV–vis) and PL spectra

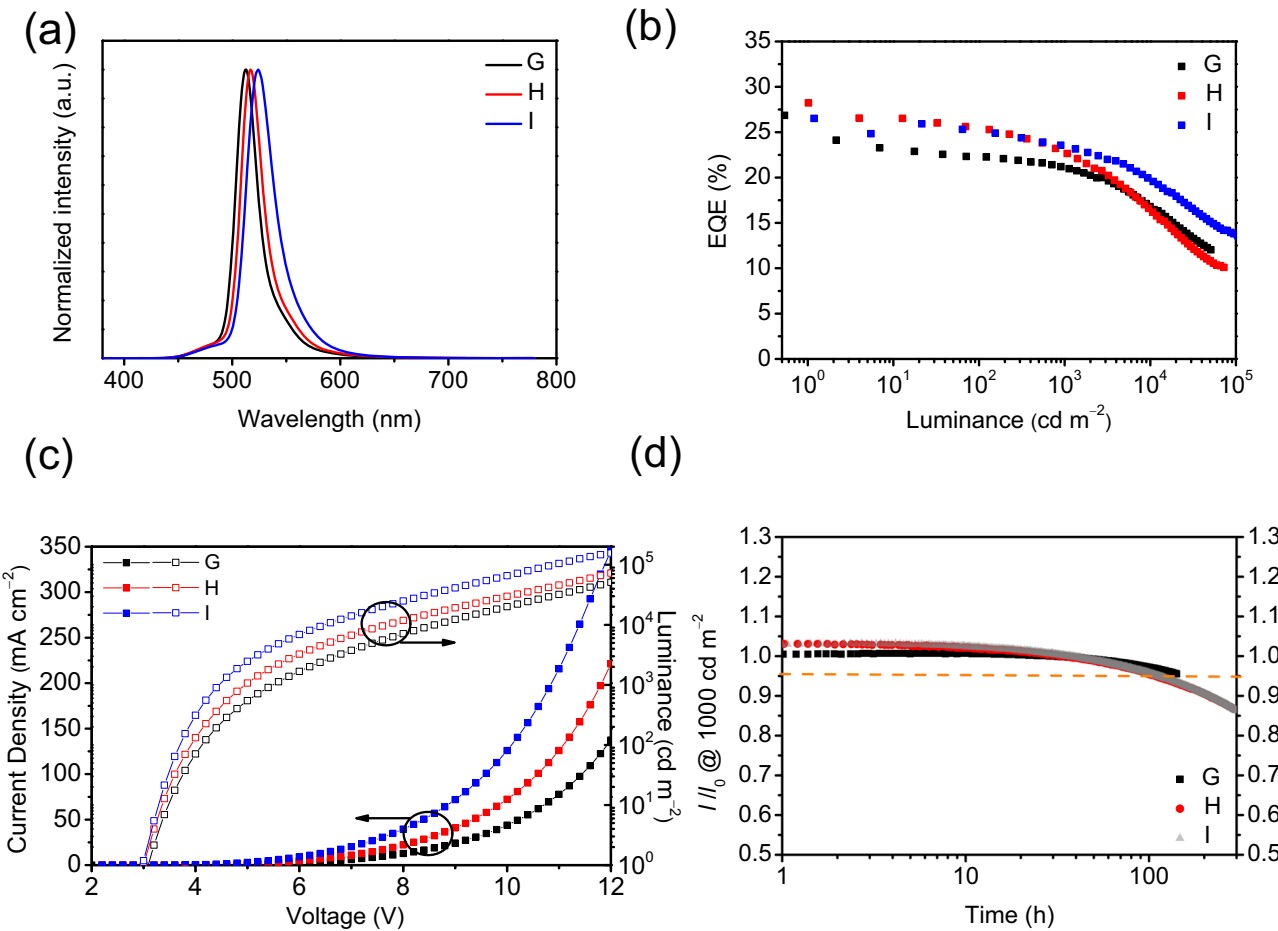

**Fig. 4 | Device performance of HF devices G – I. a** EL spectra; **b** EQE versus luminance curves; **c** current density-voltage-luminance curves and **d** device stability (at an initial luminance of 1000 cd m$^{-2}$).

were recorded on a Perkin-Elmer Lambda 950 KPA spectrophotometer and a Jobin Yvon FluoroMax-3 fluorospectrometer. Phosphorescent spectra were recorded on a JASCO FP-6500 fluorescence spectrophotometer at 77 K. Absolute PL quantum yields were measured on a Quantaurus-QY measurement system (C11347-11, Hamamatsu Photonics) under nitrogen flow and all samples were excited at 360 nm. The temperature-dependence transient decay profiles of the doped films were measured under a vacuum using a streak camera system (Hamamatsu Photonics, C4334) equipped with a cryostat (Iwatani, GASESCRT-006-2000, Japan). A nitrogen gas laser (Lasertechnik Berlin, MNL200) with an excitation wavelength of 337 nm was used. For the TrEL measurement, pulsed voltages were applied using a function generator. The emitted light was detected using a photomultiplier tube (PMT) module (H10721-01, Hamamatsu Photonics, Japan) and the current signals from the PMT were amplified using a current amplifier (DHPCA-100, Femto, Germany). All signals were measured using an oscilloscope with signal averaging performed over 1000 measurements.

### Cyclic voltammetry

Cyclic voltammetry (CV) was carried out on a CHI600 voltammetric analyzer at room temperature with a conventional three-electrode configuration consisting of a platinum disk working electrode, a platinum wire auxiliary electrode and an Ag wire pseudo-reference electrode with ferrocene−ferrocenium (Fc/Fc$^+$) as the internal standard. Argon-purged $N,N$-dimethylformamide was used as a solvent for scanning the oxidation with tetrabutylammonium

hexafluorophosphate (TBAPF$_6$) (0.1 M) as the supporting electrolyte. The cyclic voltammograms were obtained at a scan rate of 100 mV/s.

### Device fabrication and measurements

The OLEDs were fabricated by vacuum deposition process without exposure to ambient air. After fabrication, the devices were immediately encapsulated with glass lids using epoxy glue in a nitrogen-filled glove box (O$_2$ ~ 0.1 ppm, H$_2$O ~ 0.1 ppm). The indium−tin oxide surface was cleaned ultrasonically and sequentially with acetone, isopropanol and deionized water, then dried in an oven, and finally exposed to ultraviolet light and ozone for about 10 min. Organic layers were deposited at a rate of 1 Å/s. Subsequently, Liq and Al were deposited at 0.3 and 1 Å/s, respectively. The device area is ~ 0.04 cm$^2$. The EQE and $J$-$V$-$L$ measurements were performed using a Keithley 2400 source meter and an absolute external quantum efficiency (EQE) measurement system (C9920-12, Hamamatsu Photonics, Japan). For the device lifetime tests, the luminance and EL spectra of the driving devices in the normal direction were measured using a luminance meter (SR-3AR, TOPCON, Japan) under constant current density driving conditions with an initial luminance of 10$^3$ cd m$^{-2}$.

### Data availability

The data that support the findings of this study are available in the supplementary material of this article. Additional information is available from the authors on request. Source data are provided with this paper.

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

## Acknowledgements

We acknowledge Mr. Y. Kondo and Mr. R. Kawasumi (JNC Japan Co. Ltd.) for OLED fabrication, and Ms. N. Nakamura and Ms. K. Kusuhara for their technical assistance with this research. This work was supported financially by JST CREST (grant no. JPMJCR22B3), JSPS Core-to-Core Program (grant no. JPJSCCA20180005), JSPS KAKENHI (grant no. 23H05406), JSPS International Leading Research (ILR) (grant no. 23K20039), and Kyulux Inc. C.-Y. Chan thanks the financial support from the City University of Hong Kong (Project No. 9610637 and 9231531). Y.-T. Lee thanks the financial support from the National Science and Technology Council (NSTC 111-2113-M-031-008-MY3).

## Author contributions

C.-Y.C., T.H., and C.A. supervised the project. N.M. and S.U. carried out the synthesis of pure-green MREs. Y.H. carried out photophysical and electrochemical measurements. Y.-T.L., R.W.W., and G.N.I.L. fabricated the OLEDs and characterized the device performances. S.O. and M.K. carried out calculations. Y.-T.L., C.-Y.C., T.H., and C.A. contributed to the manuscript writing. Y.L. and Y.T. provided fruitful discussions. All authors discussed the progress of the research and reviewed the manuscript.

## Competing interests

The authors declare no competing interests.
