## [Peer Review File · Nature Communications]

Bright, Efficient, and Stable Pure-Green Hyperfluorescent Organic Light-Emitting Diodes by Judicious Molecular DesignREVIEWER COMMENTS

Reviewer #1 (Remarks to the Author):

As the requirements for displays become increasingly demanding, especially in terms of brightness, color purity, and operational lifetime, it is welcoming to see new progresses in the OLED field. On the basis of their previous study (ω -DABNA) in J. Am. Chem. Soc., authors tried to improve the brightness, roll-off, and above all the operational lifetime. The molecular design strategy is the peripheral modification of ω -DABNA with widely used steric groups such as methyl and phenyl, resulting in new molecules of ω -DABNA-M and ω -DABNA-PH, respectively. A complete molecular characterization and device optimization were performed and analyzed. Results show a certain degree of device performance improvement compared with their reference device (ω -DABNA with a different device structure) and an impressive device operational lifetime (LT95@1000nit \sim 600 h). I suggest publishing the work after the following key concerns are addressed.

1. NTSC and BT.2020

The authors claimed that their green-emitting devices fulfill the requirement of NTSC and BT.2020 standards, which require the CIE coordinates of (0.21, 0.71) and (0.17, 0.80), respectively. However, the CIEy values for the HF devices in Table 2 are only 0.64-0.70, which is far from BT.2020. The description here should be more accurate.

2. HOMO/LUMO levels

The experimental values of HOMOs and LUMOs are missing, which make the analysis of device performance difficult.

3. Spectral information

As to the HF devices, the spectra of sensitizer and terminal emitters should be provided for better understanding of the exciton energy transfer processes.

4. Device structure

Why the device structure of A-C is different from D-J?

5. Figure 2

(1) At low doping concentration of 0.5wt%, it is reasonable that J-V curves are identical for three emitters. But why ω -DABNA-PH shows a faster roll-off than others, especially when considering that ω -DABNA-PH has the highest kRISC?

(2) Please check the color in (d).

6. Effect of concentration

With the increase of doping concentration, the trapping density increases as well, as shown by the transient EL spectroscopy. If we look at the roll-off of the ω -DABNA-PH devices between EQE1nit and

EQE1000nit, $(EQE1nit - EQE1000nit)/EQE1nit$ increases from 9.9% in Device F (0.5 wt%) to 12.8% in Device I (1 wt%). The deteriorated roll-off results in a reduced device operation lifetime from 205 h to 109 h. So, the key issue comes to the influence of small concentration. Is it a carrier transport issue or an exciton energy transfer issue? More discussion is expected.

7. Effect of sensitizer

(1) $(EQE1nit - EQE1000nit)/EQE1nit$ of Device J increases to 24.2%. Why the operational lifetime of Device J is the longest? It is against the trend in Device F and J. Why the change of host to 4CzIPN can lead to such a boost in device operational lifetime?

(2) How about the device operational lifetime of sensitizer alone (without terminal emitters)?

8. Mistake in Table 1

Please check the PLQY values.

Reviewer #2 (Remarks to the Author):

This manuscript reports two new MR TADF emitters, ω -DABNA-M and ω -DABNA-PH, achieving the high efficiencies, brightness, stability device performances. The impact seems its high CIEy value of >0.7 near to the green standard of NTSC and BT2020, and the long lifetime of LT95 over 600 h. These impacts are very attracting and can be considered to accept after clarify the following unclear parts.

1. Due to the device stability, the spectral shift and CIE coordinates variation with increasing luminance and after aging at LT95 should be provided.
2. The reason of different efficiency roll-off in different devices should be clarified.
3. The gap between the PLQY values of MR and its HF doped film is large, which cannot reflect to the gap of their device's efficiencies. Why?
4. The reason of LT improvement should be clarified, especially device J with relative lowest efficiency and the longest LT.
5. The results of ω -DABNA-M employed to device J should be provided in SI for reference.
6. Please give the insight to explain why the low efficiency devices generally perform the longer LT.

Reviewer #3 (Remarks to the Author):

The authors synthesized and characterized two new multi-resonance TADF emitters (ω -DABNA-M and ω -DABNA-PH). In comparison with the parent molecule (ω -DABNA) the new derivatives display redshifted emission bands. The new emitters have been used to fabricate highly efficient pure-green OLEDs. The

authors have also shown that by using the hyperfluorescence strategy it is possible to significantly suppresses the efficiency roll-off and improve the brightness and color purity of the OLEDs.

I find the paper very interesting and recommend publication.

Minor comment:

Giving the Stock shifts and band full-widths in nm, strictly speaking, is ambiguous. Therefore, please also add the relevant energy values.

Reviewer #1 (Remarks to the Author):

As the requirements for displays become increasingly demanding, especially in terms of brightness, color purity, and operational lifetime, it is welcoming to see new progresses in the OLED field. On the basis of their previous study (ω -DABNA) in J. Am. Chem. Soc., authors tried to improve the brightness, roll-off, and above all the operational lifetime. The molecular design strategy is the peripheral modification of ω -DABNA with widely used steric groups such as methyl and phenyl, resulting in new molecules of ω -DABNA-M and ω -DABNA-PH, respectively. A complete molecular characterization and device optimization were performed and analyzed. Results show a certain degree of device performance improvement compared with their reference device (ω -DABNA with a different device structure) and an impressive device operational lifetime (LT95@1000nit \sim 600 h). I suggest publishing the work after the following key concerns are addressed.

A: We appreciate this strong endorsement by the reviewer.

1. NTSC and BT.2020

The authors claimed that their green-emitting devices fulfill the requirement of NTSC and BT.2020 standards, which require the CIE coordinates of (0.21, 0.71) and (0.17, 0.80), respectively. However, the CIE_y values for the HF devices in Table 2 are only 0.64-0.70, which is far from BT.2020. The description here should be more accurate.

A: We thank the reviewer for this comment. We have revised the description to be more accurate. We have revised the sentence “Herein, we report pure-green devices with CIE_{x,y} that are close to the NTSC and BT. 2020 standards.” in the abstract.

2. HOMO/LUMO levels

The experimental values of HOMOs and LUMOs are missing, which make the analysis of device performance difficult.

A: We thank the reviewer for this suggestion. We have performed the CV measurements of three MREs. From the oxidative scan of cyclic voltammograms, the HOMO energy levels of ω -DABNA, ω -DABNA-M, and ω -DABNA-PH were found to be -5.26 , -5.21 , and -5.24 eV, respectively. From the HOMO energy level and optical energy gap, the LUMO energy levels of ω -DABNA, ω -DABNA-M, and ω -DABNA-PH were estimated to be -2.82 , -2.79 and -2.82 eV, respectively. We have added the cyclic voltammogram and electrochemical section in the SI and manuscript, respectively.

Figure S27. Cyclic voltammogram of the oxidative scan of MREs in *N,N*-dimethylformamide. The HOMO energy levels of ω -DABNA, ω -DABNA-M, and ω -DABNA-PH are determined to be -5.26 , -5.21 , and -5.24 eV, respectively.

3. Spectral information

As to the HF devices, the spectra of sensitizer and terminal emitters should be provided for better understanding of the exciton energy transfer processes.

A: We thank the reviewer for this suggestion. We have included absorption spectra of terminal emitters in toluene and the emission spectra of sensitizers in doped films in the SI file, indicating excellent overlapping between sensitizers and terminal emitters. However, the low doping concentrations of terminal emitters in different HF devices result in incomplete FRET processes. Thus, the residual emissions from sensitizers were found. The residual emissions can be reduced by increasing the doping concentration of the terminal emitters.

Figure S23. Absorption spectra of MREs in toluene and emission spectra of sensitizer-only doped films.

4. Device structure

Why the device structure of A-C is different from D-J?

A: We thank the reviewer for this suggestion. Due to the shallower HOMO energy levels of MREs (ω -DABNA, ω -DABNA-M, and ω -DABNA-PH), the carrier transport properties of MREs are different from that of TADF assistant dopants. Therefore, the device structure for devices A – C was designed to maximize the EL efficiency without the assistant dopants. In HF devices, on the other hand, the major dopant is the assistant dopant. To achieve high efficiency and long stability, the device structure for devices D – K is different from those of devices A – C.

5. Figure 2

(1) At low doping concentration of 0.5wt%, it is reasonable that J-V curves are identical for three emitters. But why ω -DABNA-PH shows a faster roll-off than others, especially when considering that ω -DABNA-PH has the highest kRISC?

A: We thank the reviewer for this comment. In these devices, the dopant concentration of MRE is low and the carrier transport properties are highly dependent on MREs. Having shallower HOMO levels, devices B and C with ω -DABNA-M and ω -DABNA-PH, respectively, showed similar rolloff of ~10 %. On the other hand, device A with a deeper HOMO ω -DABNA (fewer hole traps) showed the minor rolloff issue. We believe that the HOMO and LUMO levels of MREs, which would affect the number of hole traps and the recombination zone in EML, also play an essential role in the device efficiency.

(2) Please check the color in (d).

A: We thank the reviewer for this suggestion. We have amended the color in Figure (d).

6. Effect of concentration

With the increase of doping concentration, the trapping density increases as well, as shown by the transient EL spectroscopy. If we look at the roll-off of the ω -DABNA-PH devices between EQE1nit and EQE1000nit, $(EQE1nit - EQE1000nit)/EQE1nit$ increases from 9.9% in Device F (0.5 wt%) to 12.8% in Device I (1 wt%). The deteriorated roll-off results in a reduced device operation lifetime from 205 h to 109 h. So, the key issue comes to the influence of small concentration. Is it a carrier transport issue or an exciton energy transfer issue? More discussion is expected.

A: We thank the reviewer for this comment. The triplet exciton of MRE has been found to be relatively unstable and long-lived (*Adv. Opt. Mater.* **2022**, *10*, 2200682). To minimize the direct formation of triplet excitons on MRE, the doping concentration of MRE should be low so that all excitons are generated on the assistant dopant and are

transferred to MRE via FRET. Moreover, the k_{RISC} of MRE is usually lower than that of the assistant dopant. Therefore, with an increase in the MRE concentration, more triplet excitons are directly formed on MRE, resulting in a larger efficiency rolloff. We have added the sentence of “To minimize the direct formation of triplet excitons on MRE, hence resulting in a larger efficiency rolloff.” in the manuscript.

7. Effect of sensitizer

(1) $(\text{EQE}_{1\text{nit}} - \text{EQE}_{1000\text{nit}}) / \text{EQE}_{1\text{nit}}$ of Device J increases to 24.2%. Why the operational lifetime of Device J is the longest? It is against the trend in Device F and J. Why the change of host to 4CzIPN can lead to such a boost in device operational lifetime?

A: We thank the reviewer for this comment. We believe the recombination zones were shifted depending on the assistant dopants. This may be due to the fact that 4CzIPN, with an additional cyano group, is more electron-transporting when compared to 3Cz2DPhCzBN. However, due to the faster k_{RISC} (4CzIPN: 10^{-6} s^{-1} vs 3Cz2DPhCzBN: 10^{-5} s^{-1}) and an extremely high chemical stability of 4CzIPN, the HF devices based on 4CzIPN showed extraordinary good stabilities. Moreover, we have fabricated **Ref-1** and **Ref-2** devices with the only assistant dopants. **Ref-2** with 4CzIPN showed a longer device stability than that of the 3Cz2DPhCzBN-based device (**Ref-1**).

(2) How about the device operational lifetime of sensitizer alone (without terminal emitters)?

A: We thank the reviewer for this comment. We have fabricated the sensitizer-only devices (20 wt% 3Cz2DPhCzBN:mCBP and 20% 4CzIPN:mCBP as EML). The device performance of both reference devices has been summarized in Table 2 and SI. In a 20 wt% 3Cz2DPhCzBN-based reference device, a maximum EQE of 23 % with an EL at 498 nm was achieved, corresponding to $\text{CIE}_{x,y}$ of (0.22, 0.48). The sky-blue device showed an LT_{95} of 80 h at an initial luminescence of 1000 cd cm^{-2} . On the other hand, in a 20 wt% 4CzIPN-based reference device, a maximum EQE of 21.2% with an EL of 528 nm was achieved, corresponding to $\text{CIE}_{x,y}$ of (0.33, 0.60). The green device showed an LT_{95} of 403 h. We have added the data in Figures S30 and S38 with a description in the main manuscript.

Figure S30. Device performance of TADF OLED based on 20 % 3Cz2DPhCzBN:mCBP. (a) Current density-voltage-luminance curve; (b) EQE versus luminance curve; (c) EL spectrum and (d) device stability (at an initial luminance of 1000 cd m^{-2}).

Figure S38. Device performance of TADF OLED based on 20 % 4CzIPN:mCBP. (a) $J - V$ and $L - V$ curves; (b) EQE versus luminance; (c) EL spectrum at 1000 cd m^{-2} ; (d) device stability (at an initial luminance of 1000 cd m^{-2}).

8. Mistake in Table 1

Please check the PLQY values.

A: We thank the reviewer for this comment. We have checked the PLQYs of MREs in toluene solution (10^{-5} M) and in mCBP-doped films (1 wt%). In the toluene solution, the PLQYs of ω -DABNA, ω -DABNA-M, and ω -DABNA-PH were found to be 82%, 83%, and 78%, respectively. The PLQYs found in the toluene solution are comparable to that in PMMA-doped film. However, the PLQYs of mCBP-doped films based on 1 wt% of ω -DABNA, ω -DABNA-M, and ω -DABNA-PH were found to be 57%, 60%, and 72 %, respectively. The PLQY values in 1wt% mCBP-doped film were significantly lower than those in solution and 1 wt% PMMA-doped film, while they are consistent with those of HF-doped films. We believe the lowering in PLQY originated from the host-guest interaction between MRE and mCBP. The high EQEs achieved in devices originate from the high horizontal orientation, which is commonly observed in planar DABNA-based MREs. (*Nat. Photon.* **2021**, 15, 203-207 and *Angew. Chem. Int. Ed.* **2022**, 61, e202212575)

Reviewer #2 (Remarks to the Author):

This manuscript reports two new MR TADF emitters, ω -DABNA-M and ω -DABNA-PH, achieving the high efficiencies, brightness, stability device performances. The impact seems its high CIEy value of >0.7 near to the green standard of NTSC and BT2020, and the long lifetime of LT95 over 600 h. These impacts are very attracting and can be considered to accept after clarify the following unclear parts.

A: We appreciate the strong endorsement by this reviewer.

1. Due to the device stability, the spectral shift and CIE coordinates variation with increasing luminance and after aging at LT95 should be provided.

A: We thank the reviewer for this comment. We have added the data of spectral changes of CIE_{x,y}s with increasing luminance and after aging at LT95 in Table S2 and Figures S32-35 and S39 in the SI file.

Figure S32. Change of EL spectra with respective to luminance in ω -DABNA-based HF devices (a) **D** and (b) **G**.

Figure S33. Change of EL spectra with respective to luminance in ω -DABNA-M-based HF devices (a) **E** and (b) **H**.

Figure S34. Change of EL spectra with respective to luminance in ω -DABNA-PH-based HF devices (a) **F** and (b) **I**.

Figure S35. Change of EL spectra with respect to luminance in 4CzIPN-based HF devices (a) **J** and (b) **K**.

Figure S39. Change of EL spectra with respect to time in various HF devices.

2. The reason of different efficiency roll-off in different devices should be clarified.

A: We thank the reviewer for this comment. When comparing the efficiency at 1 and 1000 cd m^{-2} in devices **D – F**, devices **D**, **E**, and **F** showed efficiency rollofts of 14.6 %, 11.7%, and 9.8%, respectively. Since the major dopant in HF devices is the assistant dopant, the carrier transport properties in HF devices mainly rely on the assistant dopant, instead of MREs. In the cases of devices **G – I**, on the other hand, it has been found that the efficiency rolloff is highly dependent on the k_{RISC} of corresponding MREs. Moreover, increasing the dopant concentration of MREs in devices **G – I**, the higher

krisc of **ω -DABNA-PH** in device **I** also resulted in the least efficiency rolloff. We have added the description in the manuscript.

3. The gap between the PLQY values of MR and its HF doped film is large, which cannot reflect to the gap of their device's efficiencies. Why?

A: We thank the reviewer for this comment. We have checked the PLQYs of MREs in toluene solution and mCBP-doped films. In the toluene solution, the PLQYs of **ω -DABNA**, **ω -DABNA-M**, and **ω -DABNA-PH** were found to be 82%, 83%, and 78%, respectively. The PLQYs found in the toluene solution are comparable to those in PMMA-doped film. However, the PLQYs of mCBP-doped films based on 1 wt% of **ω -DABNA**, **ω -DABNA-M**, and **ω -DABNA-PH** were found to be 57%, 60%, and 72 %, respectively. The PLQY values in 1wt%-doped film are significantly lower than that in solution and 1 wt% PMMA-doped film, while they are consistent with that of HF-doped films. We believe the lowering in PLQY originated from the host-guest interaction between MRE and mCBP. The high EQEs achieved in devices originate from the high horizontal orientation, which is commonly observed in planar DABNA-based MREs. (*Nat. Photon.* **2021**, 15, 203-207 and *Angew. Chem. Int. Ed.* **2022**, 61, e202212575)

4. The reason of LT improvement should be clarified, especially device J with relative lowest efficiency and the longest LT.

A: We thank the reviewer for this comment. In general, the device stability of the HF device is highly dependent on the stability of the TADF assistant dopant. In device **J**, the green TADF dopant, 4CzIPN, was used, which always displays a long device lifetime compared to the sky-blue TADF dopant (3Cz2DPhCzBN) used in other devices. As a result, even though the efficiency is lower in device J, it showed the best lifetime.

5. The results of **ω -DABNA-M** employed to device J should be provided in SI for reference.

A: We thank the reviewer for this comment. We have fabricated the device based on **ω -DABNA-M** (device **J**). The data has been added in Table 2 and SI. It has been found that device **J** results in a slightly lower maximum EQE of 19.9%, when compared to that of **Ref-2**. Although the FWHM of the EL spectrum has been improved to 26 nm with CIE_{x,y} of (0.27, 0.66), the device stability of device **J** is worse than **Ref-2**. 4CzIPN possesses a deeper HOMO level than that of 3Cz2DPhCzBN. The poorer device stability would be originated from HOMO traps, which is due to a mismatch of HOMO levels between 4CzIPN and **ω -DABNA-M**. The above description has been added to the manuscript.

6. Please give the insight to explain why the low efficiency devices generally perform the longer LT.

A: We thank the reviewer for this comment. In general, the device stability of the HF device is highly dependent on the stability of the TADF assistant dopant. In device **J**, the green TADF dopant, 4CzIPN, was used, which always displays a long device lifetime compared to the sky-blue TADF dopant (3Cz2DPhCzBN) used in other devices. Further, the degree of overlap between the absorption spectrum of the terminal emitter and the emission spectrum of the assistant dopant and the extinction coefficient of the terminal emission in the overlap range will affect the device lifetime. As a result, even though the efficiency is lower in device **J**, it showed the best lifetime.

Reviewer #3 (Remarks to the Author):

The authors synthesized and characterized two new multi-resonance TADF emitters (w-DABNA-M and w-DABNA-PH). In comparison with the parent molecule (w-DABNA) the new derivatives display redshifted emission bands. The new emitters have been used to fabricate highly efficient pure-green OLEDs. The authors have also shown that by using the hyperfluorescence strategy it is possible to significantly suppresses the efficiency roll-off and improve the brightness and color purity of the OLEDs.

I find the paper very interesting and recommend publication.

A: We appreciate the strong endorsement by this reviewer

1. Giving the Stock shifts and band full-widths in nm, strictly speaking, is ambiguous. Therefore, please also add the relevant energy values.

A: We thank the reviewer for this suggestion. We have converted the Stokes shift and FWHM into energy for a better comparison.

Tutorial

1. Since additional experiments are performed by new authors, the author list (Rangani Wathsala Weerasinghe¹, Yanmei Hu¹, Gerardus N. Iswara Lestanto¹) and author contribution section have been updated.

REVIEWERS' COMMENTS

Reviewer #1 (Remarks to the Author):

The authors have revised their manuscript on the basis of the reviewer's comments. I suggest publishing it in Nature Communications without change.

Reviewer #2 (Remarks to the Author):

The paper has been adequately revised and can be recommended for publication.